# Metabolic Adaption of Flexor Carpi Radialis to Amplexus Behavior in Asiatic Toads (*Bufo gargarizans*)

**DOI:** 10.3390/ijms241210174

**Published:** 2023-06-15

**Authors:** Chengzhi Yan, Hui Ma, Yuejun Yang, Zhiping Mi

**Affiliations:** Key Laboratory of Southwest China Wildlife Resources Conservation (Ministry of Education), China West Normal University, Nanchong 637009, China; ychzhi@cwnu.edu.cn (C.Y.); mahui102@cwnu.edu.cn (H.M.); 212021071300001@stu.cwnu.edu.cn (Y.Y.)

**Keywords:** amphibian, metabolomic, reproduction, adaption, energy demands

## Abstract

Amplexus is a type of mating behavior among toads that is essential for successful external fertilization. Most studies have primarily focused on the behavioral diversity of amplexus, and less is known regarding the metabolic changes occurring in amplectant males. The aim of this study was to compare the metabolic profiles of amplectant Asiatic toad (*Bufo gargarizans*) males in the breeding period (BP group) and the resting males in the non-breeding period (NP group). A metabolomic analysis was conducted on the flexor carpi radialis (FCR), an essential forelimb muscle responsible for clasping during courtship. A total of 66 differential metabolites were identified between the BP and NP groups, including 18 amino acids, 12 carbohydrates, and 8 lipids, and they were classified into 9 categories. Among these differential metabolites, 13 amino acids, 11 carbohydrates, and 7 lipids were significantly upregulated in the BP group compared to the NP group. In addition, a KEGG (Kyoto Encyclopedia of Genes and Genomes) enrichment analysis identified 17 significant metabolic pathways, including ABC transporters, aminoacyl-tRNA biosynthesis, arginine biosynthesis, pantothenate and CoA biosynthesis, and fructose and mannose metabolism. These results suggest that amplectant male toads are metabolically more active than those during the non-breeding period, and this metabolic adaptation increases the likelihood of reproductive success.

## 1. Introduction

Most anuran amphibians exhibit amplexus during mating [1], wherein the male approaches the female from behind and grasps her dorsally with his forelimbs to facilitate external fertilization. Several forelimb muscles are responsible for the clasping behavior during mating, such as flexor carpi radialis (FCR), abductor indicus longus, pectoralis series, triceps branchii, and extensor carpi radialis [2]. The duration of amplexus varies considerably across species, and it ranges from several hours to days or even months. The sperm ejaculated from the males fertilizes the eggs released from the females during this time or with some delay [3,4]. Mating pairs face relatively challenging conditions during the breeding season, such as prolonged clasp, reduced food intake, restricted locomotion, and competition among males, and, therefore, they have higher energetic demands than their non-breeding counterparts [5].

The metabolic requirements of reproduction in anuran amphibians show significant differences within and between the sexes at different times [6]. During the reproductive season, males expend large amounts of energy in sustained calling, mate searching, amplexus, and gamete production [7,8], while females require considerable energy to produce yolk for the eggs [9]. Energy expenditure is closely accompanied by metabolic changes. In grey tree frogs (*Hyla versicolor*), the metabolic rate of amplectant males is two-fold higher than that of resting males due to the aerobic costs of amplexus [10]. In cane toads (*Rhinella marina*), however, amplexus reduces the feeding rates and locomotor performance of females, which in turn depends on the size of the amplectant male for terrestrial but not for aquatic mating [11]. Furthermore, male tungara frogs (*Engystomops pustulosus*) in amplexus build foam nests for the eggs while grasping females, which further increases metabolic costs [12]. These reproductive behavioral changes in amplectant males depend on significant metabolic changes in the muscles involved in amplexus [13].

Although studies have been conducted to assess the energetic costs of reproduction in amphibians, the metabolic profiles of amplectant males have not been comprehensively analyzed. Mass-spectrometry-based metabolomics has emerged as a powerful tool for analyzing the metabolic changes in biological samples [14,15], and gas chromatography–mass spectrometry (GC-MS) and liquid chromatography–mass spectrometry (LC-MS) are frequently used to detect volatile and non-volatile metabolites [16]. The aim of this study was to explore the metabolic changes in amplexus-related muscles during the breeding period relative to the non-breeding period. The metabolomes of the FCR muscles of male Asiatic toads (*Bufo gargarizans*), a suitable model for evolution, behavior, and development biology, were compared [17,18]. Our findings provide new insights into the metabolic adaptation in male anuran amphibians during amplexus that maximizes the likelihood of reproductive success.

## 2. Results

### 2.1. FCR Muscle Mass Is Not Altered in Amplectant Males

During the breeding period, axillary amplexus was observed in adult toads that lasted several days (Figure 1A). Given the role of the FCR muscle in amplexus, we measured the wet muscle mass in amplectant and non-amplectant males during the breeding period (BP) and non-breeding period (NP), respectively. As shown in Figure 1B, there was no significant difference between the BP and NP groups (*p* > 0.05). Independent of body mass (BM) and body length (snout–vent length, SVL), the relative FCR wet mass (FCR wet mass/BM or FCR wet mass/SVL) was also similar in both groups (*p* > 0.05, Figure 1C,D).

### 2.2. FCR Muscle of Amplectant Males in the Breeding Period Has a Distinct Metabolic Profile

To examine the metabolic processes affected by amplexus, we compared the metabolomes of the FCR muscle of the amplectant and non-amplectant males. The distribution of the FCR samples was determined by using a principal component analysis (PCA). As shown in Figure 2A, principal components 1 and 2 had variances of 24.8% and 14.6%, respectively, indicating that the samples collected from the males during the breeding period were distinct from those collected during the non-breeding period. Thus, our experiment is reliable and reproducible.

Since the PCA model is unsupervised, we performed an orthogonal partial least squares discriminant analysis (OPLS-DA) to further distinguish the overall differences in the metabolic profiles and to screen for metabolites. As shown in Figure 2B, there was a marked separation between the two groups. In the OPLS-DA plots, R2Y (cum), Q2 (cum), and Q2 were used to evaluate the interpretation, predictive, and modeling abilities of the OPLS-DA model, and R2 values were used to indicate the corresponding goodness of fit. The model showed a high level of interpretation ability R2Y (cum), with a score of 0.997, and a good predictive ability Q2 (cum), with a score of 0.915. Moreover, the left R2 and Q2 values were smaller than the right initial values, which further confirmed the stability of the model (Figure 2C). Therefore, the variable importance of projection (VIP) from the OPLS-DA model was used to screen differential metabolites (VIP > 1). According to the PCA and OPLS-DA plots, the metabolic profiles of the FCR muscle during the breeding and non-breeding periods were distinct.

### 2.3. Identification of Differential Metabolites in the Amplectant Males

A total of 66 differential metabolites were identified between the BP and NP groups based on VIP > 1 and *p* < 0.05, of which 51 were upregulated and 15 were downregulated in the amplectant males compared to the non-amplectant males (Figure 3A). As shown in Table 1, the upregulated metabolites included 13 amino acids, peptides, and analogues; 11 carbohydrates; 7 lipids; 4 organic acids; 4 organoheterocyclic compounds; 3 organic oxygen compounds; 2 organic nitrogen compounds; and 7 other compounds. The downregulated metabolites included five amino acids, peptides, and analogues; two organic acids; two organic nitrogen compounds; and six benzenoids and other compounds. To determine the correlation between the differential metabolites, we performed a cluster analysis among ten biological repeats. As shown in the heatmap in Figure 3B, 13 out of 18 differential amino acids (e.g., L-tyrosine, beta-alanine, L-aspartic acid, 1-methylhistidine, and L-cysteine), 11/12 differential carbohydrates (e.g., L-fucose, maltotriose, D-(+)-maltose, fructose 1-phosphate, sorbitol, mannitol 1-Phosphate, and D-mannose 6-phosphate), and 7/8 differential lipids (e.g., docosahexaenoic acid, eicosapentaenoic acid, 4-hydroxybutyric acid, 2-ethylhexanoic acid, and beta-glycerophosphoric acid) were significantly more abundant in the BP group than in the NP group. These metabolites are closely associated with sugar, fat, and protein metabolism. Taken together, amplexus during the breeding period is associated with an increased synthesis and conversion of proteins, sugars, and fats, which is indicative of more active metabolism.

### 2.4. KEGG Analysis of the Differential Metabolites

The 66 differential metabolites were annotated to 52 metabolic pathways using the KEGG (Kyoto Encyclopedia of Genes and Genomes) database, of which 17 displayed significant differences (*p* < 0.05) between the 2 groups. As shown in Figure 4A, the pathways were classified into amino acid metabolism/metabolism of other amino acids (arginine biosynthesis; cysteine and methionine metabolism; glutathione metabolism; alanine, aspartate, and glutamate metabolism; etc.), carbohydrate metabolism (fructose and mannose metabolism and the pentose phosphate pathway), lipid metabolism (butanoate metabolism), the metabolism of cofactors and vitamins (pantothenate and CoA biosynthesis), nucleotide metabolism (purine metabolism), translation (aminoacyl-tRNA biosynthesis), and membrane transport (ABC transporters and neuroactive ligand–receptor interaction).

Compared to the NP group, 9 of the top 10 differential metabolites (L-aspartic acid, L-methionine, fructose 1-phosphate, gluconic acid, etc.) in the top 5 enriched pathways were increased by 1.42- to 4.47-fold in the BP group. L-glutamine, which is mainly obtained from exogenous food intake and the conversion of endogenous glucose, was significantly reduced by 4.95-fold in the BP group (*p* < 0.05, Figure 4B). Taken together, the following are enhanced in amplectant males during the breeding period: the synthesis of proteins, cofactors, and vitamins; glucose conversion; and the transmembrane transport of nutrients. Furthermore, this might be beneficial to long-lasting axillary amplexus.

## 3. Discussion

Amplexus is a crucial behavioral aspect in the reproduction of most amphibians, and it causes various physiological responses [19,20]. In the current study, we found that the FCR muscle of amplectant male *B. gargarizans* has a distinct metabolic profile during the breeding period compared to that of resting males in the non-breeding period. Based on the GC-MS platform, we identified 66 differential metabolites, of which 51 were upregulated and 15 were downregulated in the amplectant males, and we identified 17 metabolic pathways that were significantly enriched in the BP group compared with the NP group. Our results strongly suggest that the males adapt to prolonged amplexus by modulating metabolic processes.

The FCR muscle mass was not significantly different between the breeding and non-breeding periods, indicating that the metabolic differences may be primarily due to the amplectic clasping behavior itself. During the breeding season, amphibians display specific reproductive behaviors, such as calling, mate searching, amplectic clasping, and non-clasping courtship, which require additional energy [21,22]. Larger male toads can preferably engage in the above behaviors due to their resting metabolic rate being significantly higher than that of smaller toads [23]. The calling activity in frogs is supported by laryngeal muscles, and the metabolic rate during calling is 8 times higher than that in the resting period [24]. In addition, frogs with higher calling rates demonstrate elevated enzyme activity in the aerobic trunk muscles [25]. In *Bombina orientalis*, the amplectic pairs exhibit ‘fasting male–feeding female’ (FM-FF) behavior, wherein the females eat, but the males refuse any food [26]. These observations are consistent with our hypothesis that amplectic clasping behavior triggers metabolic adaptations in amphibians.

We identified 66 differential metabolites that primarily consisted of amino acids, peptides and analogues, carbohydrates, lipids, organic acids, organic nitrogen compounds, etc., most of which are involved in the metabolism of sugar, fats, and proteins and were significantly increased in the males during the breeding period. This is not surprising given that amplectant males need to be metabolically more active in order to tightly hold the females over an extended period with scarce food intake. The disruption of metabolism in Xenopus has been linked to a decrease in adult recruitment and reproductive success, ultimately affecting the development and fecundity of their offspring [27,28]. Therefore, amphibians need to maintain metabolic homeostasis to adapt to relatively harsh environments (low temperatures, drought, insufficient food, and so on) by regulating their metabolic processes [29,30,31]. For instance, amino acids and derivatives and carbohydrates are reduced in Tibetan frogs during winter, suggesting a lower metabolic rate and energy consumption due to winter stress [32]. Likewise, Moor frogs synthesize glucose and glycerol as cryoprotectants to adapt to freezing, which is accompanied by increased glycolysis and the production of lactate, ethanol, alanine, etc. [33]. Our research indicates that amplectant males have a higher demand for energy, leading to an increase in the differential metabolites involved in energy metabolism. Our and others’ previous studies have shown that changes in metabolism are closely related to physiological behavior.

Amphibians store energy in the form of sugars, proteins, or lipids, all of which can be metabolized to acetyl-coenzyme A (acetyl-CoA) to produce energy [34]. Among the differential metabolites identified in our study, 13 amino acids, 11 carbohydrates, and 7 lipids were upregulated in the males during amplexus, which is consistent with their metabolic rate being higher than that of non-breeding males. Most of the identified amino acids (e.g., L-tyrosine, beta-alanine, L-aspartic acid, 1-methylhistidine, and L-cysteine) are non-essential amino acids or metabolic intermediates, and they play important roles in nutrient absorption and metabolism, such as glucose synthesis, protein turnover, and nutrient transport and utilization [35,36]. Moreover, L-methionine and L-cysteine can also function as antioxidants [37] and reduce oxidative damage from prolonged amplexus. In winter and spring, the previously stored lipid and glycogen are mobilized by the amphibians for maintenance, gametes synthesis, and courtship behavior [38]. As a key substrate of energy metabolism, glucose can be converted from disaccharides, trisaccharides, and monosaccharides [39,40]. Consistent with this, cellobiose, L-fucose, maltotriose, D-(+)-maltose, fructose 1-phosphate, sorbitol, mannitol 1-phosphate, and D-mannose 6-phosphate were significantly enhanced in the amplectant males, which is indicative of an increase in glucose supply. Docosahexaenoic acid (DHA) and eicosapentaenoic acid (EPA) were among the differential lipids, and they are known to improve glucose homeostasis by decreasing the serum glucose level. In addition, EPA increases glucose uptake and protein accretion in skeletal muscles [41,42]. Other differential lipids, including 4-hydroxybutyric acid, 2-ethylhexanoic acid, and beta-glycerophosphoric acid, act as intermediate metabolites in gluconeogenesis. Taken together, these findings suggest that males in amplexus are metabolically more active than non-amplectant males in the non-breeding period.

Furthermore, the KEGG enrichment analysis revealed 17 significantly different metabolic pathways, and the top 5 metabolic pathways were ABC transporters, aminoacyl-tRNA biosynthesis, arginine biosynthesis, pantothenate and CoA biosynthesis, and fructose and mannose metabolism. The ABC transporters transfer a wide range of substrates, such as amino acids, lipids, sugars, and ions [43]. Aminoacyl-tRNA biosynthesis is required for the initiation of protein biosynthesis [44], and arginine biosynthesis is related to protein synthesis, the inhibition of proteolysis, and enhanced skeletal muscle mass [45,46]. The pantothenate and CoA biosynthesis pathways play an important part in the TCA cycle [47]. Fructose and mannose metabolism intermediates are used as alternative glucose sources in a low-glucose environment [48]. In our study, 9 out of the 10 most differential metabolites associated with these top 5 pathways were significantly upregulated in the amplectant males, suggesting a higher level of substrate conversion and energy metabolism. Changes in energy demands often lead to alterations in metabolites and metabolic processes. Similar to our study, the metabolic pathways in animals and humans have undergone changes to meet the energy requirements of various physiological activities, such as growth, development, and exercise [49,50,51].

## 4. Materials and Methods

### 4.1. Animals

A total of 20 sexually mature male Asiatic toads (*Bufo gargarizans*) with similar body lengths were captured at night from farmland in Nanchong City, China (30°49′ N, 106°03′ E, 251 m elevation). Ten males in amplexus were collected on January 2023 and defined as the breeding period (BP) group, and ten non-amplectant males were collected on October 2022 and defined as the non-breeding period (NP) group. All animals were euthanized via double pithing, and snout–vent lengths (SVLs) were measured to the nearest 0.1 mm using calipers. The flexor carpi radialis (FCR) from the right forelimb of each frog was quickly dissected and snap frozen in liquid nitrogen for a metabolomics analysis, and the remaining samples from the left forelimb were used for weighing, as described in previous studies [52].

### 4.2. Sample Preparation and Metabolite Extraction

The FCR samples were weighed, and 150 mg tissue of each sample was placed in a 1.5 mL Eppendorf tube with two small steel balls. The internal standard was prepared by dissolving 15 µL L-2-chlorophenylalanine (0.3 mg/mL) in methanol. The samples were extracted with 500 µL methanol/water (4/1) at −20 °C for 2 min, and then they were sonicated at 60 Hz for 2 min. After adding 120 µL chloroform, each sample was vortexed, sonicated for 30 min in an ice-water bath, and incubated for 20 min at −20 °C. The samples were then centrifuged for 15 min at 4 °C (12,000 rpm), and 500 µL of the supernatants was freeze-dried in a centrifugal dryer. All samples were pooled to prepare QC samples, and 500 µL of the supernatant was transferred to a glass vial and dried under vacuum at room temperature. To prepare the samples for GC-MS, 80 µL of methoxylamine hydrochloride in pyridine (15 mg/mL) was added to each sample, and the resulting mixture was vortexed for 3 min and incubated at 37 °C for 90 min. After adding 80 µL BSTFA (with 1% TMCS) and 20 µL n-hexane, the mixture was vortexed for 2 min and derivatized at 70 °C for 60 min. The samples were cooled to room temperature for 30 min prior to a GC-MS analysis.

### 4.3. GC-MS Analysis

An Agilent 7890B gas chromatography system combined with a 5977 AMSD system (Agilent Technologies Inc., Santa Clara, CA, USA) was used for GC-MS. The samples were separated using an ADB-5MS fused-silica capillary column (Agilent J&W Scientific, Folsom, CA, USA). Helium (>99.999%) was passed through the column at a rate of 1 mL/min. The injection volume was 1 μL, and the injector was operated in a splitless mode at 260 °C. The initial temperature of the oven was 60 °C, which was maintained for 0.5 min, and then it was gradually increased to 125 °C at a rate of 8 °C/min; to 210 °C/min at 5 °C/min; to 270 °C at 10 °C/min; and to 305 °C at 20/min, which was maintained for 5 min. The MS quadrupole and ion source (electron impact) temperatures were set to 150 °C and 230 °C, respectively. The collision energy was 70 eV. Mass spectral data were collected in full-scan mode (m/z 50–500) with the solvent delay time set to 5 min. The QC samples were injected at regular intervals (every 10 samples) during the run to assess reproducibility.

### 4.4. GC-MS Qualitative and Quantitative Analysis

The GC/MS raw data were transformed to ABF format using the ABF Converter and imported into MS-DIAL for pre-processing, including peak recognition, peak identification, MS2Decde convolution, characterization, and peak alignment. Metabolite characterization was based on the un-target database of GC-MS from Lumingbio, and a data matrix was derived. For each sample, all peak signal intensities were divided and normalized on the basis of internal standards with post-screening RSD greater than 0.3. After data normalization, redundancy removal and peak merging were performed to obtain the data matrix.

### 4.5. Statistical Analysis

A principle component analysis (PCA) of the GC-MS data was performed to monitor the overall distribution of the samples and their stability. An orthogonal partial least squares discriminant analysis (OPLS-DA) was performed to identify the different metabolites between the BP and NP groups. Model quality was assessed to avoid overfitting using 7-fold cross-validation and 200 Response Permutation Testing (RPT). In the OPLS-DA model, the variable importance of the projection (VIP) value was used to rank the overall contribution of each variable to group discrimination. A two-tailed Student’s *T*-test was used to test for significant differences in metabolites between the two groups. The screening criteria for the differential metabolites were VIP > 1 and *p* < 0.05. Differences in FCR mass (amplectant vs. non-amplectant males) were tested using one-way ANOVA with SPSS 23.0. All results are expressed as mean ± SD. *p* < 0.05 was considered statistically significant.

## 5. Conclusions

In summary, we compared the metabolomes of the FCR muscles of *B. gargarizans* males between the breeding and non-breeding periods using GC-MS. Most metabolites related to the metabolism of sugar, proteins, or lipids were significantly increased in the amplectant males, suggesting metabolic adaptation to the high energy demands of reproduction and reduced food intake during the breeding season. Our findings provide new insights into the molecular mechanisms mediating reproductive behavior in anuran amphibians.

## Figures and Tables

**Figure 1 ijms-24-10174-f001:**
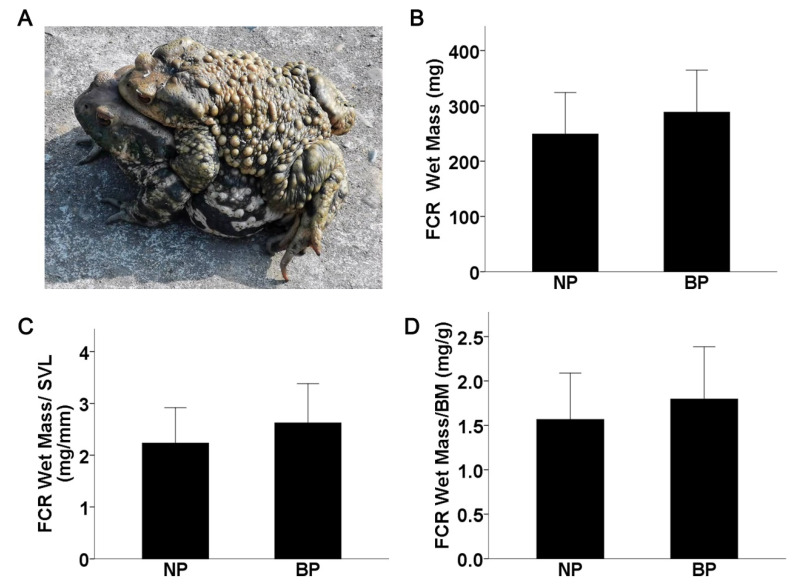
(**A**) Asiatic toads (*Bufo gargarizans*) in amplexus. (**B**) FCR wet mass comparisons between breeding period (BP) and non-breeding period (NP) groups. (**C**,**D**) The relative FCR mass (FCR wet mass/SVL and FCR wet mass/BM) comparisons between BP and NP groups.

**Figure 2 ijms-24-10174-f002:**
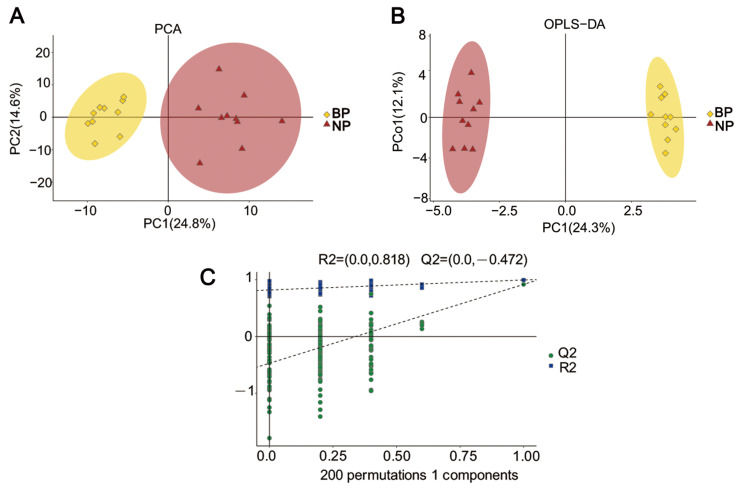
The principal component analysis (PCA) score plot (**A**), orthogonal partial least squares discriminant analysis (OPLS-DA) scores plot (**B**), and response permutation (**C**) testing for male Asiatic toads in the BP and NP groups. R2: corresponds to goodness of fit, Q2: modeling ability of the OPLS-DA model.

**Figure 3 ijms-24-10174-f003:**
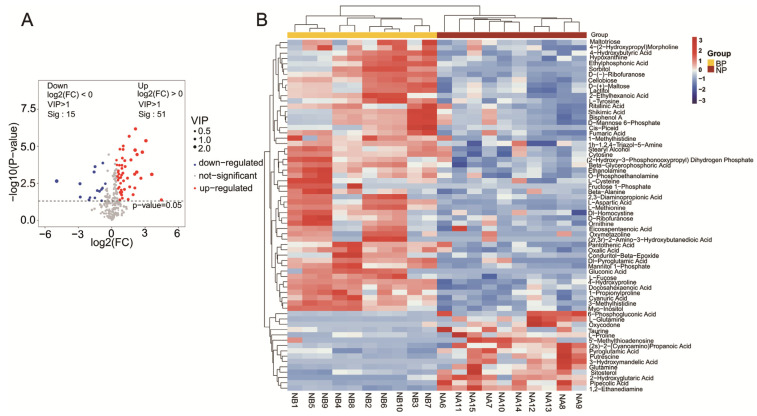
(**A**) Volcano plot of the differential metabolites of BP group versus NP group. The *p* values took the negative logarithm (−log10); thus, the ‘significance scores’ are above the dotted line (*p* < 0.05). Differential metabolites were identified between the two groups based on VIP > 1 and *p* < 0.05. The red and blue dots represent upregulated and downregulated metabolites with significant difference, respectively. (**B**) Heat map of differential metabolites in BP group and NP group. Red and blue indicate higher and lower metabolite concentrations, respectively. The differential metabolites were identified by VIP > 1 and *p* < 0.05 (n = 10).

**Figure 4 ijms-24-10174-f004:**
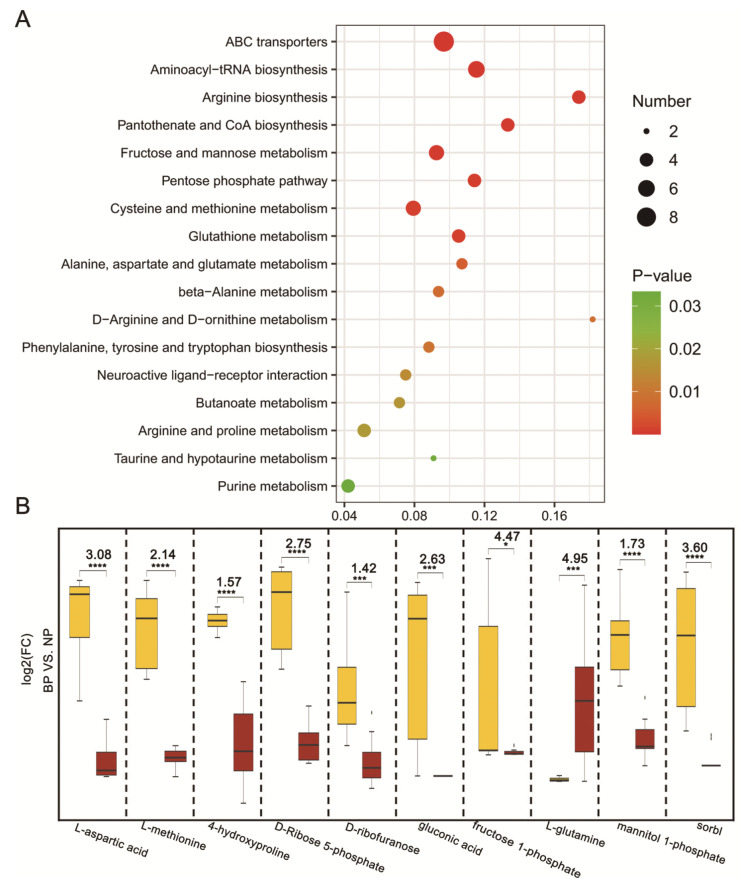
(**A**) Bubble diagram of significant enrichment pathway of BP and NP groups (*p* < 0.05). The vertical ordinate denotes metabolic pathway, and the horizontal ordinate denotes rich factor (number of significantly differential metabolites/number of total metabolites in the pathway). The size of bubbles represents the number of metabolites in the pathway. The bubble color from red to green corresponds to the *p*-value’s increase. (**B**) The top 10 differential metabolites in top 5 enrichment pathways. FC (fold change): the ratio of the average expression of metabolites in the BP and the NP groups. log2(FC): the logarithm of the ratio of the average metabolite content in the BP group (red) to the NP group (yellow). The values above asterisks represent the multiple of difference between the two groups. Significance differences are indicated by * (*p* < 0.05), *** (*p* < 0.001), and **** (*p* < 0.0001).

**Table 1 ijms-24-10174-t001:** Differential metabolites identified between the BP and NP groups.

Class	Metabolites	VIP	*p*-Value	log2(FC)	FC	Type
Amino acids, peptides, and analogues	L-aspartic Acid	2.99	4.30 × 10^−6^	3.08	8.43	Up
Amino acids, peptides, and analogues	3-methylhistidine	2.51	3.70 × 10^−5^	2.28	4.86	Up
Amino acids, peptides, and analogues	Dl-pyroglutamic acid	2.91	0.00056	2.25	4.77	Up
Amino acids, peptides, and analogues	L-methionine	2.40	6.80 × 10^−7^	2.14	4.40	Up
Amino acids, peptides, and analogues	1-methylhistidine	3.11	1.80 × 10^−5^	2.10	4.30	Up
Amino acids, peptides, and analogues	2,3-diaminopropionic acid	1.75	0.0066	2.08	4.23	Up
Amino acids, peptides, and analogues	4-hydroxyproline	1.97	2.10 × 10^−6^	1.57	2.97	Up
Amino acids, peptides, and analogues	(2r,3r)-2-amino-3-hydroxybutanedioic acid	1.06	0.01218	0.91	1.88	Up
Amino acids, peptides, and analogues	beta-alanine	1.30	0.01267	0.90	1.87	Up
Amino acids, peptides, and analogues	ornithine	1.22	0.00903	0.86	1.81	Up
Amino acids, peptides, and analogues	Dl-homocystine	1.20	0.00319	0.80	1.74	Up
Amino acids, peptides, and analogues	L-cysteine	1.11	0.00576	0.68	1.60	Up
Amino acids, peptides, and analogues	L-tyrosine	1.11	0.00045	0.60	1.52	Up
Amino acids, peptides, and analogues	pyroglutamic acid	1.11	0.0034	−0.57	0.67	Down
Amino acids, peptides, and analogues	L-proline	1.09	0.0479	−0.90	0.53	Down
Amino acids, peptides, and analogues	pipecolic acid	1.90	0.00055	−1.58	0.33	Down
Amino acids, peptides, and analogues	glutamine	2.06	0.00341	−2.85	0.14	Down
Amino acids, peptides, and analogues	L-glutamine	3.22	0.00227	−4.95	0.03	Down
Benzenoids	oxymetazoline	1.58	0.03875	1.48	2.80	Up
Benzenoids	bisphenol a	1.39	0.03241	0.96	1.94	Up
Benzenoids	3-hydroxymandelic acid	1.38	0.00989	−1.29	0.41	Down
Benzenoids	oxycodone	1.79	0.02694	−2.83	0.14	Down
Carbohydrates and carbohydrate conjugates	fructose 1-phosphate	2.10	0.04177	4.47	22.10	Up
Carbohydrates and carbohydrate conjugates	sorbitol	3.20	0.0008	3.60	12.15	Up
Carbohydrates and carbohydrate conjugates	D-(−)-ribofuranose	2.91	2.50 × 10^−5^	2.75	6.75	Up
Carbohydrates and carbohydrate conjugates	gluconic acid	2.12	0.00145	2.63	6.17	Up
Carbohydrates and carbohydrate conjugates	mannitol 1-phosphate	2.14	8.40 × 10^−5^	1.73	3.32	Up
Carbohydrates and carbohydrate conjugates	D-ribofuranose	1.82	0.00144	1.42	2.68	Up
Carbohydrates and carbohydrate conjugates	cellobiose	1.62	1.70 × 10^−5^	1.03	2.04	Up
Carbohydrates and carbohydrate conjugates	L-fucose	1.43	0.00019	0.92	1.90	Up
Carbohydrates and carbohydrate conjugates	maltotriose	1.21	0.01209	0.86	1.82	Up
Carbohydrates and carbohydrate conjugates	D-mannose 6-phosphate	1.19	0.04133	0.73	1.66	Up
Carbohydrates and carbohydrate conjugates	D-(+)-maltose	1.18	0.00139	0.66	1.58	Up
Carbohydrates and carbohydrate conjugates	6-phosphogluconic acid	1.12	0.03054	−1.51	0.35	Down
Lipids and lipid-like molecules	4-hydroxybutyric acid	1.54	0.00218	1.06	2.09	Up
Lipids and lipid-like molecules	2-ethylhexanoic acid	1.56	9.60 × 10^−6^	1.04	2.06	Up
Lipids and lipid-like molecules	stearyl alcohol	1.61	0.00065	1.01	2.01	Up
Lipids and lipid-like molecules	docosahexaenoic acid	1.56	6.30 × 10^−5^	0.96	1.94	Up
Lipids and lipid-like molecules	eicosapentaenoic acid	1.35	0.01958	0.90	1.86	Up
Lipids and lipid-like molecules	lactitol	1.40	0.00089	0.88	1.85	Up
Lipids and lipid-like molecules	beta-glycerophosphoric acid	1.15	0.00056	0.58	1.50	Up
Lipids and lipid-like molecules	sitosterol	1.29	0.0069	−0.83	0.56	Down
Organic acids and derivatives	ethylphosphonic acid	2.30	0.00077	2.07	4.18	Up
Organic acids and derivatives	o-phosphoethanolamine	1.18	0.00077	0.64	1.56	Up
Organic acids and derivatives	fumaric acid	1.15	0.00193	0.57	1.49	Up
Organic acids and derivatives	oxalic acid	1.04	8.80 × 10^−5^	0.43	1.35	Up
Organic acids and derivatives	2-hydroxyglutaric acid	1.81	0.00024	−1.50	0.35	Down
Organic acids and derivatives	taurine	1.10	0.02924	−1.95	0.26	Down
Organic nitrogen compounds	ethanolamine	1.72	9.40 × 10^−5^	1.17	2.26	Up
Organic nitrogen compounds	ritalinic acid	1.28	0.01292	0.75	1.68	Up
Organic nitrogen compounds	1,2-ethanediamine	1.39	0.00014	−0.80	0.58	Down
Organic nitrogen compounds	putrescine	1.41	0.01117	−1.10	0.47	Down
Organic oxygen compounds	myo-inositol	1.66	0.00023	1.21	2.31	Up
Organic oxygen compounds	pantothenic acid	1.74	0.00819	1.15	2.21	Up
Organic oxygen compounds	shikimic acid	1.31	0.01608	0.83	1.78	Up
Organoheterocyclic compounds	cyanuric acid	1.32	1.60 × 10^−5^	0.74	1.67	Up
Organoheterocyclic compounds	1h-1,2,4-triazol-5-amine	1.14	0.00714	0.58	1.50	Up
Organoheterocyclic compounds	cytosine	1.10	0.00108	0.52	1.44	Up
Organoheterocyclic compounds	hypoxanthine	1.02	0.00018	0.47	1.39	Up
Others	cis-piceid	1.86	0.01863	1.93	3.80	Up
Others	conduritol-beta-epoxide	1.76	0.00551	1.62	3.06	Up
Others	1-propionylproline	1.47	0.00031	0.92	1.89	Up
Others	(2-hydroxy-3-phosphonooxypropyl) dihydrogen phosphate	1.31	0.0013	0.81	1.75	Up
Others	4-(2-hydroxypropyl) morpholine	1.19	0.00019	0.69	1.61	Up
Others	5′-methylthioadenosine	1.70	0.042	−2.04	0.24	Down
Others	(2s)-2-(cyanoamino) propanoic acid	1.20	0.00836	−0.93	0.53	Down

## Data Availability

The authors confirm that the data presented in this study are available in Appendix A.

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
