# Peer review of "Metabolic Adaption of Flexor Carpi Radialis to Amplexus Behavior in Asiatic Toads (Bufo gargarizans)"

_ijms, 2023, doi:10.3390/ijms241210174_

Round 1

Reviewer 1 Report

Dear authors,

The manuscript on metabolic adaption of FCR to amplexus behavior in Asiatic toads is well written, and organized. The research was well planned and carried out, and as such should be accepted for publication. However, prior to its acceptance, I suggest correcting few minor things. Below you can find my comments:

Line 22. comparing to that àchange to “comparing to those”

Keywords: amplexus à already in the title; add some other keyword

Line 40, 41, 46, 48 – the males... the females... à delete „the“ in all cases

Line 45 - valid name for cane toad is Rhinella marina (check www.amphibiaweb.org)

Line 47 - valid name for tungara frog is Engystomops pustulosus

Apart from these, I would suggest to try to expand the discussion a bit more and to compare your data with other, similar research.  

Dear authors,

the language is good, and there is almost nothing that needs to be corrected.

Author Response

Dear reviewers, thank you for your careful review and constructive suggestions regarding our manuscript. We have revised the manuscript in accordance with the comments and marked all the amends on our revised manuscript. Our responses to the specific comments from reviewers are addressed in the attachment. Please see the attachment.

Reviewer 2 Report

The authors investigated potential differences in metabolite regulation in flexor carpi radialis muscles of Asian toads during the breeding and nonbreeding periods to infer their role in amplexus behavior. Their findings revealed important processes that drive toad metabolism during amplexus. The study is an important addition to the literature with implication for toad reproductive biology and behavior.

 Comments

Line 18: Define KEGG.

Lines 70-72: “Independ”? What did the authors mean here? They compared ratios, so they should clarify the whole sentence.

Lines 92-93 and Fig. 2C: What are RY2, R2, Q2? Please clarify. Also, R2Y is not shown in Fig. 2C.

Fig. 3A, Table 1, line 132: Please define “FC”. There is  a definition of log2(FC) in line 132, adding to confusion. Be clear. What is FC and what is log2(FC)?

Fig. 3A: The dotted line representing p = 0.05 suggests that all values with p > 0.05 are significant. I cannot interpret it otherwise. Also, grey dots are both above and below the dotted p-value line, but considered all nonsignificant. To add to the confusion, VIP thresholds are given in dot size. Please clarify, both on the graph and in its description.

Figure 4A, lines 128-129: This description is incorrect. The vertical (y) axis denotes the metabolic pathway. The horizontal axis shows p-values. The size of bubbles represents the number of metabolites in the pathway. Please amend.

Line 131: It should be increase, not decrease…

Figure 4B: Which metabolites belong to ribofuranose? Why are there two “1-phosphate”?. Please clarify.

Figure 4B: Please indicate what do the values above asterisks represent.

Line 137: Please define KEGG.

Author Response

Dear reviewers, thank you for your careful review and constructive suggestions regarding our manuscript. We have revised the manuscript in accordance with the comments and marked all the amends on our revised manuscript. Our responses to the specific comments from reviewers are addressed in the attachment .Please see the attachment.
